

# The effects of soleus fascicle length on muscle fatigability

Anh D. Nguyen[1], Aubrey Gray[1], Gregory Sawicki[2] and Jason R. Franz[1]

[1] Lampe Joint Department of Biomedical Engineering, University of North Carolina at Chapel Hill and North Carolina State University, Chapel Hill, NC, United States
[2] George W. Woodruff School of Mechanical Engineering, Georgia Institute of Technology, Atlanta, GA, United States

## ABSTRACT

**Background:** Age-related deficits in Achilles tendon stiffness have been linked to shorter calf muscle operating lengths. These shorter lengths have the potential to precipitate higher requisite excitations and thereby an earlier onset of local fatigability in older adults. However, the connection between calf muscle operating lengths and muscle fatigability has yet to be systematically explored, even in healthy younger adults. Thus, the goal of this study was to establish a mechanistic pathway between soleus fascicle length, prescribed herein using changes in ankle joint angle, and muscle fatigue in a cohort of younger adults.

**Methods:** Participants performed repeated fixed-end calf muscle contractions to task failure in a computerized robotic dynamometer at two ankle joint angles to prescribe relatively shorter (*i.e.*, 15° plantarflexion (PF15)) and longer (*i.e.*, 15° dorsiflexion (DF15)) soleus fascicle lengths.

**Results:** On average, moving from DF15 to PF15 elicited 14% shorter muscle fascicle lengths, a roughly two-fold higher EMG rms amplitude, significant but modest reductions in EMG median frequency, and a 60% reduction in time to task failure (*p*-values < 0.05). These findings may have clinical relevance for mechanisms associated with higher metabolic costs and increased fatigability among older adults, who often operate their calf muscles at shorter fascicle lengths due to lesser Achilles tendon stiffness. As a roadmap for comparison and hypothesis generation, this study sets the stage for future work in older adults.

# INTRODUCTION

Preventing walking disability in our rapidly aging population is an enormous public health challenge. In total, 13.2%, 22% and 39.4% of people aged 65–74, 75–84, and 85+ years, respectively, report that difficulty walking interferes with their daily activities (*Administration on Aging, 2023*). Unfortunately, despite considerable research efforts and clinical trial investment, the percentage of people over age 65 that report being unable to walk 2–3 city blocks has actually increased (from 18% to 21%) over the last 20 years (*Federal Interagency Forum on Aging-Related Statistics, 2024*). Fundamental to their loss of functional mobility is that, compared to younger adults, older adults exhibit a higher metabolic cost of walking (*Ortega & Farley, 2015*) and experience an earlier onset and

Corresponding author
Jason R. Franz,
jrfranz@email.unc.edu

greater magnitude of fatigue (*Lanza, Russ & Kent-Braun, 2004*; *McNeil & Rice, 2007*; *Santos et al., 2021*), which can precipitate slower walking speeds and limit physical activity. There is a critical need to better understand the underlying determinants of this higher metabolic cost and disproportionate fatigue in order to identify potentially modifiable factors for intervention.

Many factors predispose older adults to an earlier onset of muscle fatigue. Candidate factors include declines in muscle mass and strength (*Evans, 1995*), alterations in muscle fiber composition (*Lexell, 1995*), and decreased mitochondrial function (*Del Campo et al., 2018*) and capillary density (*Tonson et al., 2017*). Some growing evidence also suggests that, compared to that in younger adults, muscle fatigability is more pronounced in older adults during higher velocity *vs* isometric tasks (*Lanza, Russ & Kent-Braun, 2004*; *McNeil & Rice, 2007*). Compounding these local age-related changes in muscle tissue, we contend that changes in the operating behavior of muscle (*i.e.*, muscle fascicle length), may also precipitate an earlier onset and faster rate of muscle fatigability for older adults. However, the specific role of muscle fascicle length in governing the onset of local muscle fatigue has yet to be empirically established, even in healthy younger adults.

We (*Conway & Franz, 2020*) and others (*Mian et al., 2007*; *Panizzolo et al., 2013*) have shown that, compared to younger adults, older adults operate their calf muscles at shorter lengths which associate with a diminished capacity to enhance push-off intensity during walking (*Conway & Franz, 2020*). Conversely, younger and older adults operate with similar ankle kinematics (*Kerrigan et al., 1998*) and thereby calf muscle-tendon unit lengths (*Mian et al., 2007*). As one explanation supported by extensive prior work summarized in our recent review (*Krupenevich et al., 2022*), an age-related decrease in Achilles tendon stiffness is likely to yield shorter calf muscle lengths to generate some requisite force output—a phenomenon supported by walking simulations (*Orselli, Franz & Thelen, 2017*). According to principles of muscle physiology and the mechanics of contractile proteins, those shorter muscle lengths require higher excitations to produce a requisite force. Conceptually, those higher excitations are likely to increase the metabolic cost of producing muscle force and work, making the calf muscles prone an earlier onset of local muscle fatigue. Indeed, results from (*Beck et al., 2022*) supported this assertation, demonstrating in younger adults that shorter fascicle lengths do drive higher metabolic costs of muscle force generation.

There have been conflicting studies on the muscle length-dependence of fatigue. A longer time to task failure has been shown for operating the quadriceps (*Ng et al., 1994*) and ankle dorsiflexors (*Fitch & McComas, 1985*) at lengths shorter than optimal. Conversely, repeated stimulations to rat medial gastrocnemius muscles showed that shorter lengths elicited a faster rate of fatigue (*Macnaughton & Macintosh, 2007*). Human studies have yet to focus on the length-dependent fatigability in plantarflexor muscles during voluntary contractions and have excluded direct measures of muscle fascicle length using ultrasound.

Thus, the goal of this study was to empirically examine the effect of muscle fascicle length on the onset of local calf muscle fatigue in a cohort of young adult participants. We contend that establishing these relations in younger adults is the pivotal first step toward

developing evidence-based interventions to promote fatigue resistance and thereby walking performance for those in aging population. We focused on the calf muscles, specifically the soleus muscle, due to their relevance to walking performance and age-related mobility decline. We hypothesized that, compared to those when operating at longer fascicle lengths in a dorsiflexed position, soleus muscle activity would increase and time to task failure would decrease when performing fixed-end calf muscle contractions to failure at shorter fascicle lengths in a plantarflexed position.

## MATERIALS AND METHODS

### Participants and study design

We recruited a cohort of 16 younger adults (age: 22.7 ± 4.5 years, 10M/6F, mass: 76.0 ± 13.1 kg, height: 1.7 ± 0.1 m) who provided written, informed consent as per the University of North Carolina Biomedical Sciences Institutional Review Board (IRB #23-1615). Our sample size was determined to have 80% power to detect (*i.e.*, $p < 0.05$) a moderate effect size of 0.65 in a two-tailed pairwise comparison (*i.e.*, $n = 13$) while allowing for a conservative attrition rate of up to 20% (*i.e.*, $n = 16$ enrolled). Prior to the study, we excluded participants who reported a leg injury or fracture within the past 6 months, neurological disorder affecting the legs, were taking medications that cause dizziness, or had a leg prosthesis. In a single experimental session, participants performed a series of isolated, fixed-end plantarflexor muscle contractions to task failure while we recorded electromyographic and ultrasound data. These contractions were performed using participants' right leg while seated in a computerized robotic dynamometer (System 4 Pro; Biodex, Shirley, NY, USA) with their knee flexed to 110° (Fig. 1A) to isolate force production to the uniarticular soleus muscle (*Beck et al., 2022*). Participants feet were secured using pedal straps to minimize heel rise, which was monitored throughout the protocol. Prior to the fatiguing task described below, we collect resting *in vivo* B-mode ultrasound images of the soleus through the muscle belly of the medial gastrocnemius and aligned longitudinally with the muscle's line of action at 15° of dorsiflexion (DF15) and 15° of plantarflexion (PF15) to quantify fascicle length. Images were collected for approximately 5 s from a 60-mm linear array transducer (LV7.5/60/128Z-2; UAB Telemed, Vilnius, Lithuania) operating at 60 frames/s using an image depth of 50 cm. We quantified fascicle length from deep to superficial aponeurosis in the first image of the sequence using geometric tools in the Telemed software.

### Plantarflexor fatigue: protocol and measurements

We prepped the right leg soleus muscle belly by shaving and abrading the skin with alcohol wipes before placing surface recording electrodes (1000 Hz; Delsys, Natick, MA, USA) according to SENIAM recommendations (*Hermens et al., 2000*). Adhesive tape secured the sensor in place throughout testing. Participants then performed three maximal voluntary isometric contractions (MVIC) at a neutral ankle position (*i.e.*, 0°) from which we immediately extracted their maximum net ankle moment. Verbal encouragement were provided during each contraction. To prevent ankle joint rotation and isolate the calf muscles, we asked participants to keep their foot firmly fixed to the dynamometer pedal

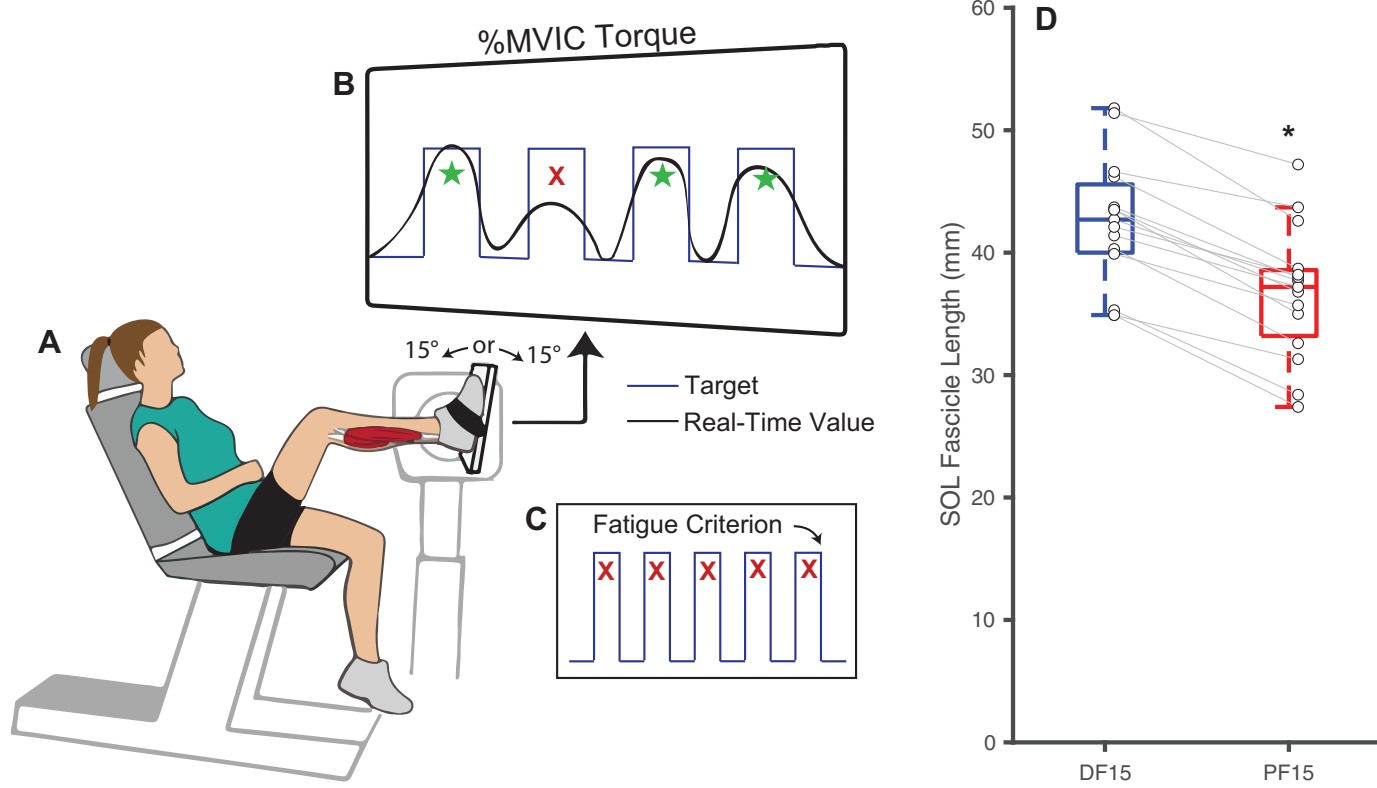

**Figure 1 Experimental methods and protocol design.** Methodological diagram showing the experimental setup and protocol design for this study. Subjects performed isolated, fixed-end plantarflexor muscle contractions to failure in a Biodex dynamometer (A) while targeting personalized square waves set to 75% of maximum voluntary isometric ankle moments (B). In randomized order and separated by a rest period, these tasks were performed at: (i) shorter muscle fascicle lengths at 15° of plantarflexion and (ii) longer muscle fascicle lengths at 15° of dorsiflexion. Participants performed the task at each angle until failing to match their target ankle moment over five consecutive pushes (failure criterion, C). (D) Shows box plots and individual subject data ($n = 16$) for ankle angle-related and between-group differences in passive soleus (SOL) fascicle lengths for 15° dorsiflexion (DF15) and 15° plantarflexion (PF15). The single asterisk (*) indicates a significant two-tailed pairwise differences between conditions ($p < 0.05$).

and secured it using Velcro straps. Thereafter, participants performed repeated fixed-end calf muscle contractions at DF15 and PF15, in randomized order and counterbalanced to prevent overperformance of one order than the other, while a real-time biofeedback system implemented in MATLAB (MathWorks, Inc., Natick, MA, USA) monitored instantaneous net ankle moment (Fig. 1B). A short one-minute trial was performed for the participant to familiarize with the biofeedback system before the official session, and verbal encouragement were provided throughout each trial. We instructed participants to push against the dynamometer pedal to replicate a continuous stream of target square waves displayed on a screen (0.5 Hz, 35% duty cycle, 100% MVIC ankle moment amplitude), resulting in a 0.7-s contraction followed by a 1.3-s rest period for each cycle. We defined the onset of fatigue as an inability to produce an average of at least 75% of that target ankle moment over 5 consecutive pushes, at which point we recorded the time to task failure (Fig. 1C). Participants were provided a 25-min rest between the two ankle angles to allow

recovery. Figure S1 displays time series data from a representative participant completing the entire experimental protocol.

## Plantarflexor fatigue: data processing and analysis

A MATLAB script (MathWorks, Inc., Natick, MA, USA) demeaned, bandpass filtered (4th-order Butterworth, 20–400 Hz), and rectified the soleus EMG signals. Dynamometer ankle moment data were filtered using a 4th-order Butterworth (15 Hz cut-off), and the resulting signal was used to define the "start" and "end" of each push using a threshold of 50% of the target value. We used these start and end times to segment the fully-conditioned EMG signals, and thereafter calculated the rms amplitude of soleus EMG for each push. We performed a similar process to extract median soleus EMG frequency from the filtered, non-rectified signals. From there, we averaged these two outcomes (rms amplitude and median frequency) for early (first 5) and late (last 5) pushes during the fatiguing task. Here, we normalized the rms amplitudes to the maximum MVIC value at the neutral ankle angle. A decrease in median frequency and an increase in normalized rms amplitude calculated from soleus EMG are used herein as neuromuscular surrogates of fatigue (De Luca, 1984; Hegedus et al., 2020; Phinyomark et al., 2012).

## Statistical analysis

After confirming normal distribution for all outcomes, we calculated descriptive statistics and, for hypothesis testing, performed one-tailed paired t-tests to compare differences between: (i) DF15 and PF15 for time of onset to task failure, calf muscle fascicle lengths, and early and late EMG median frequency and normalized rms amplitude; and (ii) early and late pushes for EMG median frequency and normalized rms amplitude. Effect sizes are reported as Cohen's d values.

## RESULTS

Changes in ankle joint angle elicited changes in passive soleus fascicle lengths (Table 1), which were 14% shorter for PF15 (36.7 ± 5.4 mm) than for DF15 (42.6 ± 5.21 mm, $p < 0.01$, d = 1.1) (Fig. 1D). Representative ultrasound images are shown in the Supplemental Material. These shorter lengths were accompanied by roughly two-fold higher EMG rms amplitudes ($p < 0.01$) (Fig. 2). Normalized rms amplitudes of the EMG signal were significantly higher in the first set of five pushes for PF15 (105.0 ± 26.2%) compared to DF15 (50.3 ± 19.0%, $p < 0.01$, d = 2.4). This heightened excitation persisted during the last set of five pushes, with PF15 exhibiting higher rms EMG (102.8 ± 23.5%) than DF15 (52.8 ± 18.6%, $p < 0.01$, d = 2.4). We found that the shorter fascicle lengths at PF15 precipitated, on average, a 60% reduction in time to task failure onset compared to the longer lengths at DF15 (200.9 ± 130.7 s and 449.5 ± 269.9 s, respectively, $p < 0.01$, d = 1.2) (Fig. 2).

Table 2 summarizes the effects of soleus fatigue on EMG outcomes. Compared to earlier contractions, later contractions exhibited lower EMG median frequency—a neuromuscular marker of fatigue. At DF15, EMG median frequencies were significantly higher in earlier pushes (106.7 ± 18.5 Hz) than later pushes (99.6 ± 19.7 Hz) ($p = 0.046$,

**Table 1 Descriptive statistics for soleus (SOL) fascicle length, time to task failure, normalized rms EMG amplitudes.**

| | Fatigue onset (s) | SOL length (mm) | Normalized RMS EMG (%) | |
| --- | --- | --- | --- | --- |
| | | | Early | Late |
| DF15 | 449.5 ± 269.9 | 42.6 ± 5.2 | 50.3 ± 18.9 | 52.8 ± 18.6 |
| PF15 | 200.9 ± 130.7 | 36.7 ± 5.4 | 105.0 ± 26.2 | 102.8 ± 23.5 |
| *p*-value | <0.001 | <0.001 | <0.001 | <0.001 |

Note:
DF15, 15° dorsiflexion; PF15, 15° plantarflexion.

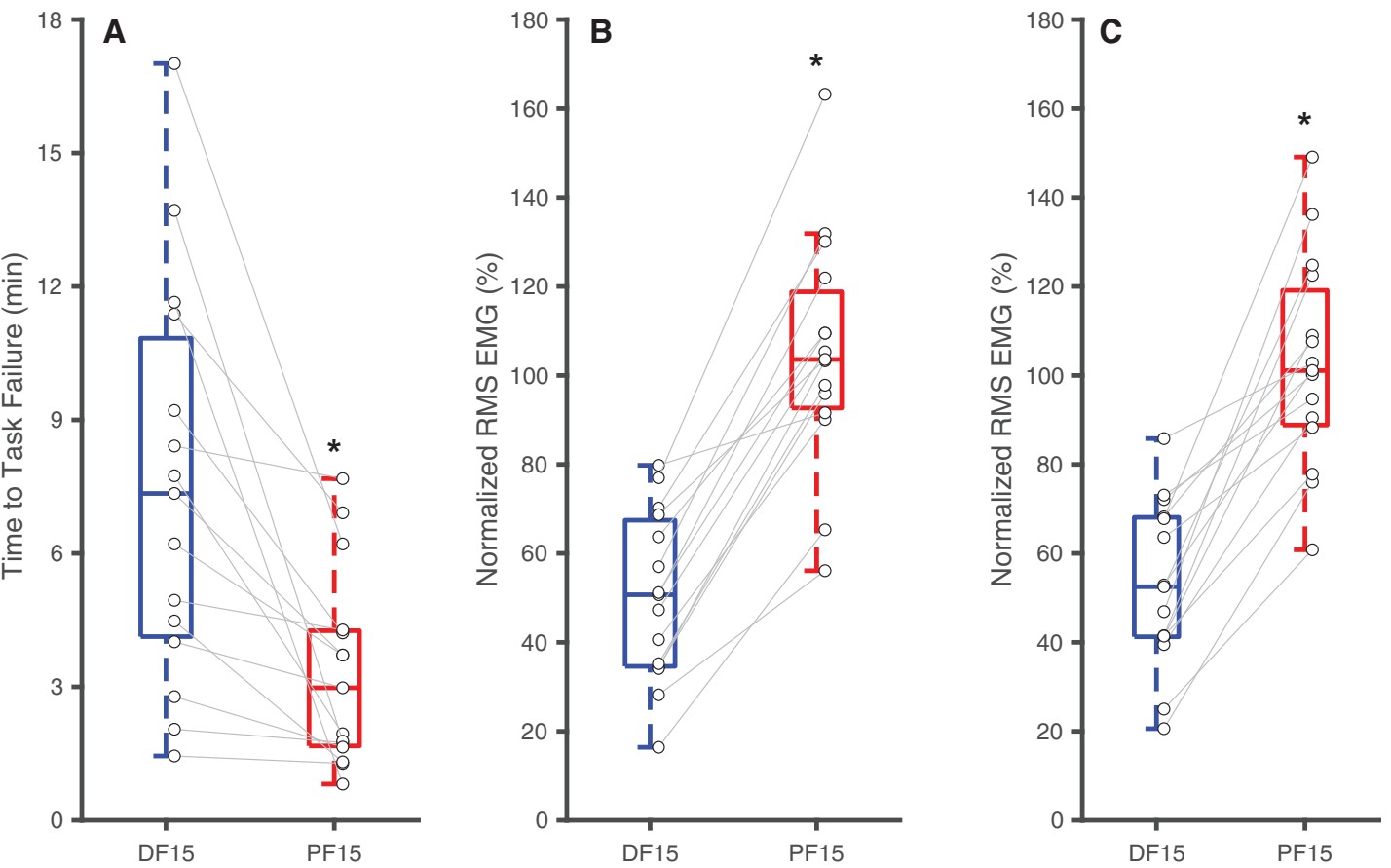

**Figure 2 Differences in time to task failure and normalized EMG amplitudes.** Box plots and individual subject data (*n* = 16) showing ankle angle-related differences in time to task failure (A), normalized EMG amplitudes for earlier pushes (B), and normalized EMG amplitude for later pushes (C) for 15° dorsiflexion (DF15) and 15° plantarflexion (PF15). Asterisks (*) indicate significant two-tailed pairwise differences between conditions (*p* < 0.05).

d = 0.37) (Fig. 3A). This pattern persisted at PF15, where EMG frequencies remained higher in earlier (150.1 ± 17.3 Hz) than later pushes (140.8 ± 20.8 Hz) (*p* < 0.01, d = 0.48) (Fig. 3B).

**Table 2 Descriptive statistics for EMG median frequency.**

| | EMG frequency (Hz) | |
| --- | --- | --- |
| | **DF15** | **PF15** |
| Early | 106.7 ± 18.5 | 150.1 ± 17.3 |
| Late | 99.6 ± 19.7 | 140.8 ± 20.8 |
| *p*-value | 0.046 | 0.007 |

**Note:**
DF15, 15° dorsiflexion; PF15, 15° plantarflexion.

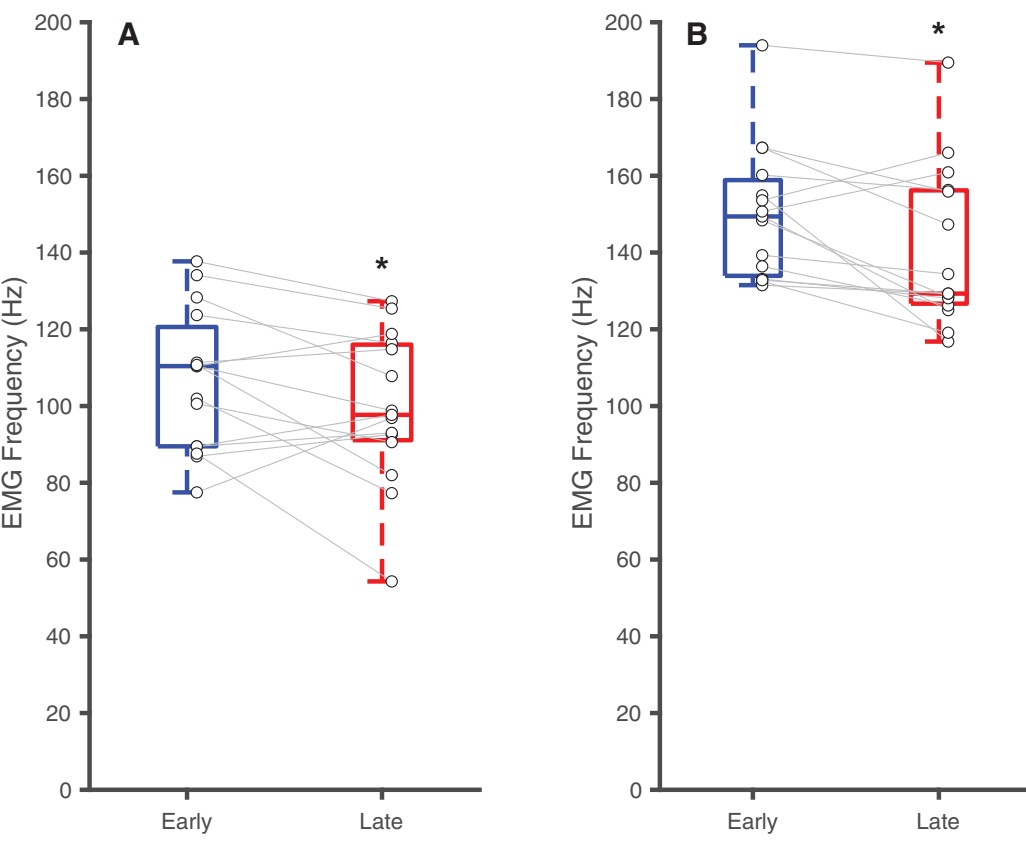

**Figure 3 Differences in EMG median frequency.** Box plots and individual subject data (*n* = 16) showing ankle angle-related differences in EMG median frequency at (A) 15° dorsiflexion and at (B) 15° plantarflexion for earlier and later pushes. Asterisks (*) indicate significant two-tailed pairwise differences between conditions (*p* < 0.05).

## DISCUSSION

Our purpose was to empirically establish the effects of soleus muscle fascicle length on muscle activity and the onset of muscle fatigue in a cohort of younger adults. Our cumulative findings underscore the significant influence of muscle fascicle length on local muscle fatigability of the calf muscles, which contribute in significant and meaningful ways to locomotor performance and economy and are disproportionately afflicted by aging and disease. As hypothesized, compared to longer lengths, shorter calf muscle fascicle lengths

prescribed in a plantarflexed position necessitated higher levels of excitation to meet a requisite force outcome, which consequently led to an earlier onset of fatigue. This result aligns with fundamental principles of muscle physiology and, as we describe in more detail below, may provide a roadmap for hypothesizing the relevance of our discoveries for populations that operate their calf muscle fascicles at shorter lengths such as older adults and individuals with certain diseases.

Our objective fatigue criterion was an inability to generate a requisite ankle plantarflexor moment for five consecutive contractions—a criterion met by design for all participants. In addition, the median frequency of EMG signals can serve as an indicator of fatigue, a conclusion based on the progressive loss of force-generating capacity from faster-conducting and more susceptible muscle fibers and has been established extensively in prior work (*De Luca, 1984*; *Hegedus et al., 2020*; *Phinyomark et al., 2012*). Consistent with this phenomenon, we noted a decrease in EMG median frequency during later contractions independent of ankle joint angle. Thus, we interpret our subjects as having reached a threshold of meaningful fatigue at both ankle angles based on measured objective and neuromuscular markers.

Muscles have an optimal length at which overlap between actin and myosin monofilaments and thereby force generating capacity is maximized. At shorter muscles lengths, and thereby with fewer binding sites available for those contractile proteins, higher excitations are necessary to produce a requisite force—an assertion supported by experimental data in human subjects (*Beck et al., 2022*). Conceptually, those higher excitations should precipitate an earlier onset of local muscle fatigue. Thus, given this theoretical basis, our results are not entirely unexpected. We first documented that, independent of time, operating the calf muscles at PF15 required larger EMG rms amplitudes than at DF15. Consequently, we also revealed a 60% earlier onset of calf muscle fatigue for the shorter muscle operating lengths prescribed at PF15 than the longer muscle operating lengths prescribed at DF15.

The findings of this work may have important translational implications. As one example, age and age-related declines in physical activity can significantly impact muscle function and contribute to mobility impairment. As summarized in our recent review (*Krupenevich et al., 2022*), most *in vivo* measurements in human subjects support the conclusion that aging decreases Achilles tendon tissue stiffness. A similar mechanical outcome is observed with disuse that accompanies decreased physical activity (*Alfredson et al., 1998*; *Magnusson & Kjaer, 2019*; *McCrum et al., 2018*). As a pivotal consequence, older adults are forced to operate their calf muscles at shorter lengths during functional activities such as walking (*Conway & Franz, 2020*; *Mian et al., 2007*; *Panizzolo et al., 2013*). These effects would be independent from but perhaps exacerbated by age-related declines in muscle size and strength. Certainly, additional work at the individual sarcomere level and context to optimal lengths is warranted. Nevertheless, our findings that shorter muscle fascicle lengths have the potential to accelerate calf muscle fatigue are particularly relevant. Personalized rehabilitation strategies that focus on enhancing muscle endurance, perhaps more specifically through augmenting muscle length and/or tendon mechanical properties (*i.e.*, increasing Achilles tendon stiffness), could promote fatigue resistance and thereby

enhance walking performance. As an alternative, the use of assistive devices such as exoskeletons can augmented the structural stiffness of the calf muscle-tendon units and precipitate longer muscle operating lengths during walking (*Krupenevich et al., 2022*; *Nuckols & Sawicki, 2020*). There may be an opportunity to view exoskeletons, for example, through the lens of fatigue resistance devices to enhance walking endurance.

There are several limitations to our study. First, although we discuss the implications of our results in the context of walking performance and mobility in older adults, we focused solely on otherwise healthy younger subjects, and our results may fail to generalize. Second, our study design included a 25-min rest break between the two ankle angles, the order of these angles was randomized, and subjects anecdotally described beyond sufficiently rested prior to starting the second fatiguing task. However, this rest break may not have allowed for identical recovery for all participants and may have added to between-subject variation. Third, we did not measure EMG on the quadriceps and hamstring muscles, and therefore cannot fully rule out their contributions to the MVICs and the net ankle moment. However, we believe that proper dynamometer alignment and subject postures mitigated these effects. In addition, although we secured subjects' feet and monitored throughout the trial, there may have been unanticipated heel-lift during contractions that affected the torque measurements and muscle operating lengths. Fourth, we normalized EMG amplitudes to a single MVIC value taken at a neutral ankle joint angle, which may be subject to differences in angle-torque-EMG relations. Fifth, we used 2D ultrasound to capture calf muscle fascicle lengths using a straight-line assumption. These can be subject to errors due to changes in fascicle spatial alignment. Though, given the fixed, passive joint rotations measured here, we suspect these errors are negligible. Sixth, our chosen ankle angles were intended to examine length-related effects rather than emulate anticipated aging-related differences. Thus, the 14% shorter soleus fascicle lengths reported here exceed the 7–10% decrease adopted by older adults (*Conway & Franz, 2020*; *Mian et al., 2007*; *Panizzolo et al., 2013*). Future studies should test smaller angle differences that reflect aging-related changes to better assess the ecological relevance and sensitivity of fatigability outcomes. We also fully interpret joint angle-related differences in our outcome measures in the context of muscle fascicle length differences. However, we must acknowledge also that joint angle can alter, for example, EMG signals themselves and/or muscle-tendon moment arms spanning the ankle and thereby the mechanical advantage of those muscles to generate force independent of length. Further studies, to include musculoskeletal modeling efforts, would add valuable insight to disentangle those effects. Finally, our interpretations exclude a more nuanced discussion of the potential for parallel elastic contributions to muscle force output at longer lengths. However, prior work, including that during walking, suggests that calf muscle sarcomeres at both ankle positions studied here are likely operating on the ascending limb of their force-length curve (*Hessel et al., 2020*; *Rubenson et al., 2012*).

## CONCLUSIONS

Our results point to a functional consequence of the increased physiological demand placed on calf muscles when operating at shorter lengths—namely accelerated fatigue. We

contend that these findings have clinical relevance for mechanisms associated with mobility impairment among older adults, who often operate their calf muscles at shorter lengths. These insights may have practical implications for personalized rehabilitation and training protocols designed to allow the calf muscles to operate at longer lengths (either by directly targeting the muscle itself or indirectly by increasing tendon stiffness) and may help refine ways—such as assistive devices—to mitigate fatigue in older adults.

## ACKNOWLEDGEMENTS

We thank Andrew Shelton for assistance with EMG processing.

### Funding

This work was supported by a grant from the National Institutes of Health (R01AG058615). There was no additional external funding received for this study. The funders had no role in study design, data collection and analysis, decision to publish, or preparation of the manuscript.

### Grant Disclosures

The following grant information was disclosed by the authors:
National Institutes of Health: R01AG058615.

### Competing Interests

The authors declare that they have no competing interests.

### Author Contributions

- Anh D. Nguyen conceived and designed the experiments, performed the experiments, analyzed the data, prepared figures and/or tables, authored or reviewed drafts of the article, and approved the final draft.
- Aubrey Gray conceived and designed the experiments, analyzed the data, authored or reviewed drafts of the article, and approved the final draft.
- Gregory Sawicki conceived and designed the experiments, authored or reviewed drafts of the article, and approved the final draft.
- Jason R. Franz conceived and designed the experiments, prepared figures and/or tables, authored or reviewed drafts of the article, and approved the final draft.

### Human Ethics

The following information was supplied relating to ethical approvals (*i.e.*, approving body and any reference numbers):

The University of North Carolina Biomedical Sciences Institutional Review Board approved this study (IRB #23-1615).

### Data Availability

Raw data is available in the Supplemental Files.

## Supplemental Information

Supplemental information for this article can be found online at http://dx.doi.org/10.7717/peerj.19842#supplemental-information.

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
