# Peer review of "The effects of soleus fascicle length on muscle fatigability"

_PeerJ, doi:10.7717/peerj.19842_

## Round 0.1 · original submission · Major Revisions

This study is well-structured and executed, with reasonably straightforward revisions suggested by the reviewers. The clear theoretical framework linking fascicle length and force generation is the basis for the expected relationship between posture and fatigue. The relevance of experimental demonstration of this theoretical concept with specific measurements is relevant for further exploration and comparisons in the elderly population, where the increased compliance of tendons has been demonstrated. Since this is the general rationale of this submission, the reporting and discussion should reflect this goal. The reviewers have strongly commented on the need to connect the goals of this study to the findings of this study.

Consequently, describing the logic for selecting two test postures relative to the force-length curves and the expected shift of these relationships in older adults would be beneficial.

The discussion of results from young participants should preserve the value of these findings for the examination of the elderly population. For example, examining the limitations of testing fatigue and clarifying the implications for testing in older adults might be beneficial.

Please see specific and general comments from the reviewers for the specific details.

Reviewer 1 ·

Basic reporting

Line 66: Authors state that this metric has not improved in 20 years, but cite a 13-year old source. Are there more recent data to support this claim?

Experimental design

Line 136: It would be helpful to state that the order was both randomized and counterbalanced to prevent over performance of one order than the other within your sample.

Lines 139-140: This sentence is a concern with regard to my overall comment addressing ankle position. If MVC was determined in neutral position and fatigability was assessed in plantar- or dorsiflexed positions, subjects enjoy a much greater mechanical advantage in dorsiflexion which will yield longer times to fatigue, as less force is required to meet the demand of 75% MVC. This is related to fascicle length but also to mechanical advantage of the calcaneal lever at the ankle joint. This difference in positioning provides an advantage in force production that may not be representative of the difference between young and old adults. It would be helpful to see the effects of multiple positions on these tasks so that, even if this is overestimating the difference, we can see the spectrum of times to fatigue throughout these different positions and draw conclusions about what they may mean for fatigability. The current data only tell us about the “extremes” and do not separate mechanical advantage from fascicle length.

Validity of the findings

The authors state in their introduction (line 85) that metabolic cost has been observed to be greater for older adults than young adults and this may be due to reduced Achilles tendon stiffness in aging. However, they also note that older and young adults do not display different gait kinematics. And while modifying the position of the foot segment relative to the shank is the only way to produce fascicle shortening/lengthening in young adults, it is not clear that a 30-degree difference in ankle position is representative of the potential difference in fascicle length between young and old adults due to reduced Achilles tendon stiffness. This makes the positions of 15 degrees of dorsiflexion and plantarflexion seem arbitrary.

These notes considered, it is my opinion that the authors should add these considerations to their limitations, contextualize their findings with regard to what literature does exist related to the extent of fascicle length differences between older and young adults, and comment on the potential of future research to address this issue by evaluating fatigability at multiple different ankle positions in young adults.


Lines 188-190: This is an overstatement of your findings. Data collected in young adults may allow us to hypothesize about what differences may exist between old and young adults, but certainly cannot provide a mechanistic understanding for their causes.

Additional comments

This work seeks to provide an initial step toward better understanding the effects of altered calf muscle fascicle length in older adults on fatigability by first understanding its effects on fatigability in young adults. This manuscript is well-written, with great attention to detail in its methods to facilitate repeatability. While the manuscript does seek to address an important question within the field, it is my opinion that its discussion and conclusions require more context due to the study’s limitations.

Reviewer 2 ·

Basic reporting

Overall, the manuscript is well-written and constructed. The figures are informative. I provided some comments on the literature.

Experimental design

Research is within the scope of the journal. The knowledge gap is identified in the introduction and the experiments seem to work towards addressing it. The methodology currently falls short because some technical details are needed to make the data reproducible. After inclusion, I think the experimental design is sound.

Validity of the findings

All OK.

Additional comments

I provide my more traditional comments below, which I think will improve the manuscript:

I reviewed the paper “The effects of calf muscle fascicle length on local muscle fatiguability”, and overall it is a sound research paper. The authors are trying to identify the underlying mechanism responsible for age-associated increases in muscle fatiguability. Among other possibilities, older individuals have more compliant Achilles tendons, and so during contraction, this may lead to more muscle shortening that shorts the muscle operating range onto the ascending limb of the force-length relationship. This means that it costs more energy per unit force the muscle generates and so can reasonably lead to faster muscle fatigue vs. younger adults with stiffer tendons activating at relatively longer fascicle lengths. Their study had young participants perform repeated maximum plantarflexions until failure at 2 angle joint angles (to simulate short and long fascicle lengths) while measuring ankle joint torque, fascicle length, and EMG. They report that, indeed, time to task failure was faster when fascicles were activated at the shorter fascicle length (more plantarflexed).

Introduction
The first paragraph sets up the problem well and provides plenty of background information.

Line 80: “However, the specific role of muscle fascicle length in governing..” This sentence is jarring only because fascicle length has not yet been directly discussed above. I suggest bringing it up as an example of the “operating behavior of muscle” in the previous sentence. Or remove “However”, and make it the first sentence of the next paragraph.

Line 90-91: Shorter fascicle lengths could be less efficient if their sarcomeres are on the ascending limb of the FL curve, the fascicle pennation angle is changed, etc. I would add some references that indicate where the fascicles are on the force-length relationship, during walking. There are several studies that used B-mode ultrasound on the topic. 2 examples are Rubenson et al 2012 doi: 10.1242/jeb.070466, and Hessel et al. 2021 doi: 10.1242/jeb.235614.

Line 99-100: This is not accurate (see citations above) unless I am misinterpreting this sentence. Please revise.

Materials and Methods
General comments: Please indicate if a familiarized session was used. Please indicate if motivation was used when asking participants to produce max contraction (cheering them on / yelling at this). Please add the ultrasound analysis method and stats used on those parameters.

Line 122: Please explain or cite the placement/orientation of the ultrasound probe.

Line 130: Please explain or cite the placement / orientation of the EMG electrodes.

Line141-142: What was the pause / rest period between each push during the fatigue trial? Maybe it can be worked out from details in line 139, but maybe lay it out in seconds?

Discussion
General comments.
It can be non-trivial to explain EMG signals because changes in joint angle will also change the underlying muscle structure and volume below it also impact the EMG signal, regardless of neural drive. Please discuss this effect and the ways you combated it in your discussion.
In my experience, conducting multiple MVCs in a plantarflexed position can lead to muscle discomfort and cramping from the contraction at short fascicle lengths. Did participants complain about discomfort, or is it possible that this effect also impacted their fatigue score?

Generally, I am happy to see the authors putting effort into linking their findings to their broader significance.

Line 242: Ecological? This sentence is a bit unclear.

Reviewer 3 ·

Basic reporting

Thank you for the opportunity to review this manuscript. I apologise to the authors for the delay in providing my feedback. The manuscript is well written, and I appreciate the effort invested in the study design and analysis. However, I have some concerns regarding the rationale presented in the Introduction, which may not be fully appropriate for the population tested. I recommend that the authors revise this section for clarity and relevance.
The main hypothesis of this study is based on the premise that the population is ageing rapidly, becoming less fit, and experiencing reduced functional mobility, leading to an increased metabolic cost during various activities. Investigating the underlying mechanisms is valuable, particularly given the limited research on how muscle mechanics change with fatigue and whether fatigue is fascicle-length dependent. However, it is unclear how these concerns relate to the healthy young individuals tested in this study. If the aim is to address an ageing-related issue, how does this study contribute without directly examining older adults? This argument may be better suited to the Discussion rather than as a justification for conducting the study. I would appreciate clarification from the authors on this point.
Additionally, I have methodological concerns. The study aims to examine the role of muscle fascicle length in the onset of muscle fatigue. However, to do so rigorously, participants should have been positioned at the same relative muscle length (e.g., in relation to optimal length, Lo) rather than at standardised joint angles. Given the significant variability in resting fascicle length across individuals, as well as the variable changes in fascicle length with joint angle adjustments, it is unclear whether the study truly examined the role of fascicle length in fatigue. It is not evident whether the chosen joint angles resulted in fascicles operating at different functional lengths (e.g., shorter vs. longer than optimal). As such, the study primarily investigates electromyographic (EMG) characteristics in response to fatiguing exercises at two different muscle-tendon unit (MTU) lengths rather than directly assessing the effect of fascicle length on fatigue.
Furthermore, the lack of reporting on fascicle behaviour during the fatiguing task precludes any mechanistic explanation or link between fascicle behaviour and fatigue. This may be further confounded by individual differences in tendon stiffness.

Below are specific comments on sections requiring further clarification:
Title:
Is it necessary to specify "local muscle fatigue"? While this likely refers to peripheral fatigue within the muscle, surface EMG is influenced by higher-order neural factors (e.g., spinal and cortical contributions). Without additional measures of central fatigue, this phrasing may be misleading.
Consider specifying that the study examines the effects of soleus fascicle length on muscle fatigability.
Abstract:
The first sentence would be clearer if split into two sentences.
The background is well written but does not provide a clear rationale for studying healthy young adults.
Line 37: Remove "critical."
Line 38: Explicitly state that the tested muscle is part of the calf.
The final background statements suggest a focus on ageing populations, but the study does not investigate older adults. The rationale should be revised accordingly.
Line 41: The study manipulated joint angles, not posture—the posture remained unchanged.
Line 42: The ankle was rotated rather than moved. Additionally, the study used fixed-end contractions, meaning the ankle was placed at two different joint angles, which resulted in differences in fascicle length.
Line 43: Is a 14% change in fascicle length sufficient to alter the fascicle’s operating range on the force-length curve (i.e., ascending vs. descending limb)?
Lines 45-48: Please specify the clinical relevance of these findings and how they may impact rehabilitation practices.
Given that older adults were not tested, can these findings be extrapolated to ageing populations?
Introduction:
Line 80: This argument is strong enough to justify the study without referencing ageing-related factors.
Lines 101-102: While the physiological and mechanical rationale for this study is clear, it is unclear how this relates to older adults, given that the study examined healthy young individuals. Large sections of the Introduction discussing ageing may be unnecessary.
Line 105: Did the study examine all calf muscles? Be specific about which muscle(s) were analysed.
Line 106: Rephrase for clarity—e.g., "The conditions compared were plantarflexed vs. dorsiflexed."
Methods:
Line 113: Why were 16 participants chosen? Was an a priori power analysis conducted?
Line 121: Cite a reference—does positioning the knee at 120° effectively isolate the soleus, or could the gastrocnemius still contribute? Were electrodes placed to confirm minimal gastrocnemius activation?
Lines 121-124: A representative ultrasound image would be helpful. How was fascicle length measured? Was fascicle curvature accounted for, or was it assumed to be a straight line?
Line 134: Maximal isometric contractions at these positions can lead to heel lift, which may change as the foot becomes more compliant during exercise. Was this monitored over time and between conditions?
Line 141: How was randomisation performed? Was it balanced so that half the participants started in PF15 and half in DF15?
Were any recovery tests conducted to ensure participants fully regained force within 25 minutes?
Line 150: What does "calculated the EMG for each push" mean? Was only the up-and-hold phase considered, or was the downward phase included as well?
Lines 152-153: Why was EMG normalised to the MVIC at a neutral ankle angle? Would it not be more appropriate to normalise to the MVIC at the specific joint angle used, to account for the angle-torque-EMG relationship?
Lines 154-155: Provide a reference.
Statistical Analysis:
The data appear skewed—are t-tests appropriate for this dataset?
Lines 160-162: I did not find these correlations in the Results section—please clarify.
If the data are not normally distributed, consider using non-parametric correlations or data transformation.
Results:
The ankle joint angle was changed, but the posture remained the same—please ensure consistency in terminology throughout the manuscript.
Lines 173-174: Please revise for clarity—"Soleus fatigue was also associated with hallmark changes in EMG independent" is unclear.
Line 177: A p-value of 0.046 is very close to the significance threshold—consider reporting mean differences and confidence intervals, and discuss their impact in the Discussion.
Discussion:
Line 207: Be cautious with terminology—you have not directly measured muscle length.
Line 221: Please clarify what is meant by “augmenting tendon mechanical properties.”
Lines 222-223: Is there evidence supporting this statement?
Conclusions:
Lines 245-247: Clearly state the practical implications of these findings.

Experimental design

See comments above

Validity of the findings

See comments above

Additional comments

See comments above

---

## Round 0.2 · accepted · Accept

Thank you for the final clarifications.

Reviewer 2 ·

Basic reporting

After this recent revision, I believe the paper meets requirements for basic reporting.

Experimental design

The manuscript now meets the requirements of experimental design.

Validity of the findings

The revised versions meet the requirements of this section.